# Structural and Functional Changes and Possible Molecular Mechanisms in Aged Skin

**DOI:** 10.3390/ijms222212489

**Published:** 2021-11-19

**Authors:** Hyunji Lee, Yongjun Hong, Miri Kim

**Affiliations:** Department of Dermatology, Yeouido St. Mary’s Hospital, College of Medicine, The Catholic University of Korea, #10, 63-ro, Yeongdeungpo-gu, Seoul 07345, Korea; o0or5r5r@gmail.com (H.L.); yongjun2yo@hanmail.net (Y.H.)

**Keywords:** skin aging, intrinsic aging, photoaging, molecular mechanisms

## Abstract

Skin aging is a complex process influenced by intrinsic and extrinsic factors. Together, these factors affect the structure and function of the epidermis and dermis. Histologically, aging skin typically shows epidermal atrophy due to decreased cell numbers. The dermis of aged skin shows decreased numbers of mast cells and fibroblasts. Fibroblast senescence contributes to skin aging by secreting a senescence-associated secretory phenotype, which decreases proliferation by impairing the release of essential growth factors and enhancing degradation of the extracellular matrix through activation of matrix metalloproteinases (MMPs). Several molecular mechanisms affect skin aging including telomere shortening, oxidative stress and MMP, cytokines, autophagic control, microRNAs, and the microbiome. Accumulating evidence on the molecular mechanisms of skin aging has provided clinicians with a wide range of therapeutic targets for treating aging skin.

## 1. Introduction

Skin is the outermost organ of the human body, with an extensive surface area of 1.5–2 m^2^ [1]. Skin aging, resulting in cumulative changes in skin structure, function, and appearance, is a complex process affected by both intrinsic and extrinsic factors. Skin aging is not only a physiologic phenomenon but also a health risk, resulting in increased skin fragility, delayed and impaired wound healing, and increased incidence of infection and skin cancers. Intrinsic or chronological skin aging can be seen in areas unexposed to sunlight, revealing the influence of genetic factors. Photoaging, also referred to as extrinsic aging, mainly results from ultraviolet (UV) irradiation, and mainly occurs on the face and forearms due to frequent exposure to sunlight, and other factors such as air pollution and cigarette smoke [2,3].

Generally, alterations in skin structure, function, and appearance are more pronounced in photoaged than chronologically aged skin. However, these two types of aging are difficult to separate [4] and are superimposed in sun-exposed skin, as they have common clinical features caused by dermal matrix alterations that contribute to wrinkle formation, laxity, and fragility [5]. The dermal matrix contains extracellular matrix (ECM) proteins such as collagen, elastin, and proteoglycans that confer skin strength and resilience. Skin aging associated with dermal matrix alterations and atrophy can be caused by the senescence of dermal cells such as fibroblasts, and decreased synthesis and accelerated breakdown of dermal collagen fibers [3]. In the last 30 years, substantial progress has been made in understanding the molecular mechanisms responsible for the aging of human skin. This review focuses on the molecular mechanisms of skin aging and summarizes its clinical and histological features.

## 2. Histology of Chronological Aging and Photoaging

Histological changes in chronologically aged skin are characterized by decreased recovery capacity, altered permeability of the stratum corneum, epidermal atrophy particularly affecting the stratum spinosum, and decreased amounts of fibroblasts and collagen in the dermis (Figure 1). The stratum corneum consists of corneocytes, which are anulceated keratinocytes retaining keratin filaments within a filaggrin matrix and surrounded by a cornified lipid envelope; they organize into a brick-and-mortar formation within the ECM, forming lipid-rich membranes [6]. A recent biochemical study using cadaveric donors found that the stratum corneum, especially the keratin fibers in chronologically aged skin, stiffens with aging. The kinetics of water movement through the stratum corneum is decreased [7]. Another study showed that the thickness, intercellular organization of lipids, and concentration of natural moisturizing factors of the stratum corneum increase with aging [8]. These results suggest that decreased transepidermal water loss with aging might be caused by a thickened and stiffened stratum corneum, along with increased cellular cohesion, due to altered lipid structure. The epidermal atrophy of chronologically aged skin, which occurs especially in the stratum spinosum, is related to a decreased epidermal turnover rate due to prolonged cell cycles [9]. Several studies of skin aging have reported that the epidermis thins due to decreased numbers of melanocytes, mast cells, and Langerhans cells. The number of melanocytes decreases by 8–20% each decade [10]. Additionally, the number of Langerhans cells in the epidermis markedly decreases, with noticeable morphological alterations and functional impairment. Langerhans cells dramatically decrease in density in both intrinsically and extrinsically aging skin [11]. Confocal microscopy has shown that dermal papillae and thickness of the basal layer decrease, resulting in flattening of the dermal–epidermal junction [12].

The dermis of chronologically aged skin has fewer mast cells and fibroblasts. Dermal fibroblasts are responsible for the synthesis and organization of the ECM in every tissue and organ of the body, including the skin. Electron microscopy studies have shown changes in the structure and morphology of fibroblasts from the dermis of aged skin [9]. An emerging hypothesis is that these changes, referred to as “fibroblast senescence”, are the main cause of irreversible skin aging due to their inherent ability to be almost free from apoptosis and to not be eliminated by the adaptive immune system [13]. Fibroblast senescence contributes to skin aging by secreting a senescence-associated secretory phenotype (SASP), which decreases proliferation by impairing the release of essential growth factors and enhances degradation of the ECM by activating matrix metalloproteinases (MMPs) [14]. Recently, Basisty et al. [15] presented the SASP Atlas, which is a proteomic database of soluble proteins and exosomal cargo SASP factors from multiple senescence inducers and cell types. The authors further suggested several possible biomarkers of the aging process, such as growth/differentiation factor 15, stanniocalcin 1, and serine protease inhibitors.

Collagen and elastin, two of the main constituents of the ECM, change throughout the life cycle. In the dermis of chronologically aged skin, density, thickness, and the degree of organization of dermal collagen, which are the main contributors to overall tissue stiffness and resilience, are all decreased [11]. Zouboulis et al. [16] showed that collagen synthesis sharply decreases after menopause in chronologically aged skin. Varani et al. [17] showed that, in chronologically aged skin, dermal collagen production decreases with aging, and the amount of collagen in people aged ≥80 years is 75% lower than in young adults. A negative feedback loop in the collagen synthesis process in aged skin has been hypothesized to inhibit collagen synthesis through high-molecular weight collagen fragments [15]. Additionally, collagen fragmentation by MMPs increases oxidative levels within damaged cells, which contributes to skin aging and fibroblast damage [16]. In the dermis, collagen reduction lowers the mechanical tension on fibroblasts and efficiency of collagen synthesis compared with younger skin [17]. Collagen degradation leads to loss of skin stiffness and resiliency, which manifest clinically as wrinkling and sagging [18]. The superficial dermis and dermal papillae of the skin also contain microvessel structures. Although UV radiation induces angiogenesis, cutaneous blood vessels decrease in number, size, and architectural complexity in photoaged and chronologically aged skin [19].

Elastic fibers, which are composed of filaments called microfibrils and an amorphous component, show structural changes during the aging process. In the papillary dermis, fibrillin-rich microfibrils are selectively disintegrated during chronological aging. Furthermore, fibulin-5 (FLBN5) binds to tropoelastin, a precursor of elastin, or FLBN1, as a scaffold protein to organize elastic fibers [20,21]. FLBN5 is associated with elastic fiber remodeling in the dermis during chronological aging [22,23]. In the skin of young people, FLBN5 is localized within elastic fibers and distributed throughout the dermis; however, in photoprotected aged skin, this FLBN5 expression is absent. In summary, with aging, the ECM structure is decreased and disrupted, which is correlated with functional changes in aged skin such as loss of elasticity and wrinkles [24].

Photoaged skin is characterized by several histologic findings distinct from those of chronologic aging skin. In photoaged skin, the thickness and composition of the epidermal rete become heterogeneous [25]. The thickness of the epidermal rete of photoaged skin is greater than that of chronologic aged skin, whereas epidermal atrophy is observed in severely photoaged skin [26]. Additionally, atypical melanocytes and keratinocytes can be increased by UV radiation [27]. Melanogenesis is upregulated for neutralization of reactive oxygen species (ROS) induced by UV radiation exposure and can act as a mechanism protecting against photodamage [28]. In this context, a cadaveric study found that darker skin is more photoprotected than fair skin because of the increased melanin [29].

The most notable histological feature of photoaging is the abundance of pathologically altered elastic fibers, commonly referred to “solar elastosis” [30]. Altered elastic fibers, which are commonly amorphous, thickened, curled, and fragmented, are essentially nonfunctional. Elastotic material consists of elastin, fibrillin, and glycosaminoglycans, particularly hyaluronic acid and versican (a large chondroitin sulfate proteoglycan). The pathogenesis of solar elastosis could be the result of both degradation and de novo synthesis of elastic fibers due to UV exposure, although the mechanism is not understood fully [31]. The expression level of FLBN5, an important constituent of elastic fiber associated with tropoelastin and microfibrils, is markedly decreased with UV irradiation. Paradoxically, in solar elastosis, FLBN5 is increased with other elastic fiber components [32]. The main cause of elastic fiber structure destruction in photoaging is activation of MMPs, which also characterizes the degradation of collagen fiber [33]. The enzymes responsible for the breakdown of elastic fibers are MMPs 2, 3, 9, 12, and 13 [34]. Among them, MMP12, also known as human macrophage metalloelastase, plays the most important role in elastic fiber degradation [35]. Upon UV radiation, expression of the MMP12 gene is elevated and MMP12 protein activity is increased, resulting in solar elastosis [36]. Moreover, Imokawa et al. [37] reported that repetitive UV radiation induces elevated skin fibroblast-derived elastase activity and destruction of elastic fiber structure, leading to subsequent loss of elasticity, appearing as wrinkling and sagging skin. In photoaged skin, not only the density of elastic fibers, but also the synthesis of new elastic fibers, are reduced. Cenizo et al. [38] showed that the messenger RNA (mRNA) expression of lysyl oxidase and lysyl oxidase-like, rather than the expression of elastin mRNAs, is responsible for elastin crosslinking. These mRNA levels are decreased with aging, resulting in elastogenesis inefficiency [38]. Another study reported that, in photodamaged skin, alternative splicing of the elastin gene occurs, leading to inadequate synthesis of the proteins required for correct assembly of elastic fibers [39,40]. Additionally, the increased hyperplastic/activated fibroblast and MMP levels induced by UV irradiation can lead to the synthesis of glycosaminoglycans and proteoglycans, another major constituent of the ECM [41]. Moreover, inflammatory cells such as eosinophils, lymphocytes, mast cells, and mononuclear cells are increased in photoaging skin [42]. Finally, the amount of ECM decreases and the breakdown of collagen fibers increases [43]. Various studies related to histological findings of chronological aging and photoaging are described in Table 1.

## 3. Molecular Mechanisms of Skin Aging

### 3.1. Telomere Shortening

Telomeres, consisting of repetitive TTAGGG sequences, are located at the ends of linear chromosomes in eukaryotes. Together with other proteins, telomeres constitute the shelterin complex. At the end of the chromosome, telomeres play a role in inhibiting degradation or fusion with the surrounding chromosome through the recognition of double-stranded breaks [45]. Telomere shortening indicates dysfunction of this protective mechanism, resulting in cellular senescence and aging via activation of the DNA damage response (DDR) [46]. Telomerase is responsible for adding the telomeric sequence TTAGGG to the 3′ end of telomeres to maintain telomere length [47]. Telomeres play a significant role in chronological skin aging. Dysfunctional telomeres activate the DDR pathway, which in turn activates downstream effectors such as the cyclin-dependent kinase inhibitors p16 and p21 for cell-cycle arrest. Gradual shortening of telomeres can explain the cellular senescence caused by intrinsic aging because it results in cell division [48].

On the other hand, UV exposure induces telomere shortening by producing ROS in the skin [5,49]. Oikawa et al. [50] demonstrated that exposure to UVA induces site-specific DNA damage in the telomere sequence, leading to telomere dysfunction. By contrast, Sugimoto et al. [51] reported that telomere length is shortened with time, and there is no difference between sun-exposed and sun-protected areas in this respect [52]. The molecular events and functions of telomere shortening, particularly as they pertain to skin aging, are not fully understood. However, because the advances in research in this field are all recent, further important advances are expected shortly.

### 3.2. Oxidative Stress and MMPs

In the past 30 years, a considerable number of studies have been conducted on the correlation between skin aging and oxidative stress and/or MMPs. ROS are accumulated by free radicals, which are indispensable for mitochondrial aerobic metabolism and are considered a main factor in chronological aging. Mitochondrial dysfunction is a major effector in both chronological aging and photoaging and may link the two phenomena [50,51]). According to previous studies, about 1.5–5% of oxygen is converted to ROS in sun-protected areas of the skin [53]. Similarly, in photoaging, UV exposure can cause accumulation of ROS and production of nitric oxide, resulting in skin inflammation and wrinkle formation [54,55]. Oxidative stress, which is defined as an oxidant–antioxidant imbalance, is induced by ROS accumulation [56]. When oxidative stress accumulates in cells, membrane phospholipids can be oxidized, which leads to disruption of the transmembrane signaling pathway [57].

In fibroblasts of the dermal skin, ROS accumulation results in DNA damage [58]. Additionally, excessive ROS activates the two main regulatory signaling pathways for SASP by senescent cells, the mitogen-activated protein kinase (MAPK) and nuclear factor kappa B (NF-κB) signaling pathways, which induce activator protein-1 (AP-1) for the expression of c-Fos, c-Jun, and NF-κB [14,59,60]. Activated AP-1 and NF-κB result in increased tumor necrosis factor-α (TNFα) and MMP expression. MMPs are a superfamily of zinc-containing metalloproteinases that have the capacity to degrade the ECM molecules that comprise the skin dermal connective tissue [61]. In particular, induction of AP-1 is elevated in MMP1 (collagenase), MMP3 (stromelysin-1), and MMP9 (92 kDa gelatinase), resulting in the degradation of ECM components in human skin in vivo [59]. The combined actions of MMP1, MMP3, and MMP9 decompose type-I and type-III dermal collagen into fragmented, disorganized fibrils. These decomposed products downregulate collagen synthesis, suggesting a negative feedback loop in collagen synthesis via collagen breakdown [60]. Furthermore, AP-1 activated by the MAPK pathway induces Smad7, which blocks Smad2/3 via the transforming growth factor beta (TGF-β) receptor, thereby regulating TGF-β signaling and inhibiting collagen production by dermal fibroblasts and reducing collagen density [62,63].

Increased ROS also results in the oxidation of macromolecules, such as cellular lipids, proteins, and DNA, which cause cellular dysfunction with aging. Oxidative protein damage is the main biomarker of aging, and is frequently observed in photodamaged skin [64]. Cellular accumulation of lipofuscin, a large protein–lipid aggregate consisting of 30–70% proteins and 20–50% lipids, gradually increases with age, often forming “age spots”. Lipofuscin acts as another redox-active site for free radicals and proteasome inhibitors, eventually initiating a cycle leading to protein aggregation [57]. Figure 2 summarized the major signaling pathways involved in the aging process.

### 3.3. Cytokines in Aging Skin

Cytokines are another important element in the skin aging process [65]. Upon UV radiation, several inflammatory signaling pathways are activated via different surface receptors such as epidermal growth factor receptors, the TGF receptor, Toll-like receptors, the interleukin 1 (IL-1) receptor, and the TNF receptor [66]. Major cytokines secreted from keratinocytes are interleukins (IL-1, IL-3, IL-6, IL-8, IL-33), colony-stimulating factors (CSFs) (granulocyte macrophage [GM]-CSF, M-CSF, G-CSF), TGF-α, TGF-β, TNF-α, and platelet-derived growth factor [67]. UV radiation can activate signaling directly via ROS production, or indirectly through DNA or mitochondrial damage, which causes inflammation. It is characterized by increased levels of circulating inflammatory cytokines and a shift towards cellular senescence, and these changes are known to cause many age-related diseases, including dementia, arthritis, and type 2 diabetes [68].

TNF-α, a main effector cytokine in proinflammatory processes of the skin, inhibits collagen synthesis and induces MMP9 elevation, as mentioned in a previous section. When exposed to persistent TNF-α, the production of MMP-9 is disturbed, and the epidermis can be damaged irreversibly [69]. Increased TNF-α level is associated with multiple pathways including NF-kB, AP-1, hypoxia-inducible factor 1-alpha (HIF-1a), and nuclear factor erythroid 2-related factor 2 (Nrf-2) [70]. Increased TNF-α level is associated with multiple pathways, including NF-kB, AP-1, hypoxia-inducible factor 1-alpha (HIF-1a), and nuclear factor erythroid 2-related factor 2 (Nrf-2), which are associated with MMP up-regulation [70,71].

The level of IL-1 increases with age and promotes skin inflammation, which induces age-related processes [72]. Upon UV radiation, the IL-1 receptor antagonist (IL-1ra), a competitive inhibitor of IL-1, is stimulated and plays a regulatory role in the IL-1-related proinflammatory response. IL-1ra production in the skin decreases with age, while IL-1α levels were higher in aged skin [72]. In chronological aging and photoaging processes, IL-1 and IL-6 induce the activation of key transcription factors associated with inflammatory and immune responses, such as NF-κB, AP-1, c-Jun N-terminal kinase (JNK), and MAPKs [73,74,75]. Additionally, IL-18 is an IL-1 superfamily cytokine that acts as an angiogenic mediator in inflammatory processes. In aged skin, heterodimerization of IL-18 receptor activates the NF-κB signaling pathway; IL-18 is associated with the pathogenesis of several age-related diseases, acting as a strong proinflammatory mediator by inducing interferon-γ [76]. Another proinflammatory cytokine, IL-6, is elevated in elderly people and is related to skin aging [77,78]. Kim et al. [79] showed that, in female skin after menopause, IL-6 expression is increased and associated with wrinkle formation. The IL-6 level is increased by exposure to UV radiation, indicating its association with photoaging [79,80].

Cysteine-rich protein 61 (CCN1), an ECM-associated matricellular protein, can be induced in dermal fibroblasts through AP-1-dependent activation [81]. Increased CCN1 expression is observed in the dermal fibroblasts of aged human skin [82]. CCN1 stimulates the production of MMP1, thereby contributing to significant type-I collagen degradation in the dermis [82]. Furthermore, CCN1 downregulates TGF-β type-II receptor, inhibiting the TGF-β signaling essential for maintaining ECM homeostasis [83,84].

### 3.4. Autophagic Control

Autophagy (ATG) is an evolutionarily conserved mechanism of cell self-digestion for removing unwanted or damaged proteins, lipids, and other cellular constituents by delivering these materials to the lysosome for degradation. ATG consists of five main stages: initiation, nucleation, elongation, fusion, and cargo degradation. The detailed molecular mechanisms of ATG are extensively reviewed in Klionsky et al. [85]. The key protein in the initiation phase, mammalian target of rapamycin (mTOR) C1, acts in concert with other serine-threonine protein kinases. mTORC1 is inhibited by cellular and environmental stresses incompatible with continued growth, such as glucose and amino acid deprivation, DNA damage, and hypoxia [86]. These signals activate the Unc-51-like kinase 1 complex, which induces phosphorylation of the components of the class III phosphoinositide 3-kinase complex I, thus enabling nucleation of the phagophore. The elongation of autophagosomes requires the ubiquitin-like conjugation system to orchestrate the activity of ATG-related proteins, microtubule-associated protein light chain 3 (LC3) and/or GABA type A receptor-associated protein (GABARAP) [87]. The ATG12–ATG5–ATG16L1 complex enhances the final connection of phosphatidylethanolamine (PE) molecules, resulting in the formation of membrane-bound LC3-II and/or GABARAP-PE [88]. Cellular membranes contribute to elongation of the phagophore by providing membrane material, which gives rise to double-layered vesicles called autophagosomes. UV radiation resistance-associated gene (UVRAG) competes with ATG14L for binding to Beclin 1. When bound to Beclin 1, UVRAG stimulates RAB7 GTPase activity and autophagosome fusion with lysosomes [89]. Autophagosome-lysosome fusion is managed by syntaxin 17 on autophagosomes, vesicle-associated membrane protein 8 on lysosomes, and accessory proteins such as ATG14 and homotypic fusion, and the protein sorting homotypic fusion and HOPS (homotypic fusion and vacuole protein sorting) tethering complex [90].

The formation stage involves various proteins that form autophagosomes. In the degradation stage, autophagosomes and lysosomes fuse to form autolysosomes [91]. ATG acts on damaged mitochondria marked with ubiquitin Ser65 [92] and readily aggregated cytoplasmic proteins [93]. ATG relieves intracellular oxidative stress by degrading oxidized proteins and lipids, while also contributing to many physiological and pathological activities of organisms such as tumor development and regulation, cell survival and death, and aging [94,95].

Autophagic processes significantly extend the replicative lifespan of individual cells and inhibit stress-induced cellular senescence in vivo. Furthermore, inhibition of ATG might result in premature senescence [96]. The senescent cell count and SASP secretion level are elevated after treatment with tobacco smoke extract and ATG inhibition in primary bronchial epithelial cells. However, cigarette smoke extract-induced cellular senescence is significantly inhibited by rapamycin, an mTOR inhibitor, to activate the autophagic process [97].

Despite the above research, the association between ATG and skin aging remains ambiguous. A significant increase in autophagic vacuoles has been observed by electron microscopy in senescent fibroblasts [98] and senescent keratinocytes [99]. A study by Young et al. [100] showed that fibroblast senescence might depend on prior ATG, and pharmacological or genetic methods for reducing ATG can slow the senescence process. However, some studies drew different conclusions on how ATG contributes to the aging process or showed that aging and ATG are independent processes [56,101].

Upon knockdown of the ATG-associated proteins ATG7 and ATG12, or lysosomal-associated membrane proteins in human primary fibroblasts, premature aging has been observed in a ROS- and p53-dependent manner, suggesting that inhibition of ATG is associated with senescence [102]. DNA damage and aging are abnormally increased when ATG-deficient keratinocytes are exposed to oxidative stress [100]. Thus, although ATG might be required for UV-mediated senescence, this assumption requires validation under other UV-exposure conditions, and in other cell types [103].

### 3.5. Apoptosis in Skin Aging

Apoptosis promotes tissue integrity by removing injured cells without evoking inflammation. However, apoptosis seems to be a double-edged sword; during low-level chronic stress, as seen in aging, for example, increased resistance to apoptosis can lead to the survival of functionally deficient, post-mitotic cells with poor housekeeping functions. Aging cells are markedly resistant to apoptosis, and several studies have indicated that host defense mechanisms can enhance anti-apoptotic signaling, which subsequently induces a senescent, pro-inflammatory phenotype during the aging process [104]. At the molecular level, age-related resistance to apoptosis mainly results from a functional deficiency in the p53 network.

In physiological conditions, DNA damage activates checkpoint pathways and DNA repair mechanisms. Checkpoint proteins such as p53 and ataxia-telangiectasia mutated (ATM) are activated upon UV exposure and provoke cell cycle arrest to allow proper DNA repair [105]. A gene downstream of ATM, p53, is a transcription factor that primarily functions as a gatekeeper for DNA mutations [106]. During aging, the efficiency of the stress recognition system in the ER declines [107], which can prevent the initiation of apoptosis. The functional activity of p53 has been shown to decline during aging in a murine model [108,109]. It has also been demonstrated that premature aging induced by forced activation of p53 does not represent a physiological aging process. The functional inefficiency of p53 could explain the reduction of mitochondrial respiration and increase in glycolytic metabolism seen during aging [110].

### 3.6. Role of microRNAs in Skin Aging

MicroRNAs (miRNAs) are small non-coding RNAs that post-transcriptionally regulate mRNA translation and are involved in most biological and pathological processes, including aging. The expression of specific miRNAs can serve as a biomarker of aging, including chronological aging and photoaging, as well as a marker of age-related diseases [111]. The role of miRNAs in skin aging is related to the regulation of molecules involved in the insulin-like growth factor 1 and mTOR signaling pathways [112].

Analyses of age-associated cutaneous miRNAs in keratinocytes by Rivetti et al. [113] showed increased expression of miR-130, miR-138, and miR-181a/b during replicative senescence; these processes target the p63 (a member of the p53 superfamily involved in epidermal development and tumor regulation) and sirtuin 1 (SIRT1) (a NAD-dependent histone deacetylase involved in cellular differentiation, metabolism, immune response, and apoptosis) mRNAs. Upregulation of miR-137 and miR-668 is associated with β-galactosidase activity, along elevated levels of p16INK4A and p53 senescence markers of the ARF/p53 and p16INK4A/RB pathways, respectively [114]. In addition, miR-191 is capable of blocking the G1–S phase transition. This blockade is manifested as cell cycle arrest and a quiescent state that contributes to the development of senescence processes [115].

The downregulation of miRNAs is associated with the reduced expression of transmembrane receptors and components of the ECM in senescent skin fibroblasts. Senescent dermal fibroblasts show increased levels of miRNAs, such as let-7d-5p, let-7e-5p, miR-23a-3p, miR-34a-5p, miR-125a-3p/5p, miR-152, miR-181a-5p, and miR-221/222-3p, which affects all phases of the cellular life cycle [116]. MiR-152 can significantly decrease dermal fibroblast adhesion through the downregulation of integrin alpha 5, which participates in cell-surface-mediated signaling [115]. The expression of collagen XVI, a minor component of the cutaneous ECM, in senescent fibroblasts is downregulated directly by upregulated miR-181a [113]. Additionally, the miR-29 and miR-30 miRNA families are upregulated during fibroblast senescence, which directly regulates the expression of the B-Myb transcription factor present in all cell cycle phases [117]. Additionally, the miR-17-92 cluster and miR-106 are downregulated in aged dermal fibroblasts, resulting in activation of the p53, ERB, and MAPK signaling pathways [118]. In Langerhans cells, overexpression of miR-449 and miR-9 is related to the downregulation of key molecules in the TGF-β signaling pathway, resulting in decreased function of Langerhans cells [119].

Upon UV irradiation, miRNA expression is changed; miR-27a, miR-145, miR-383, and miR-1246 levels are increased, and mi-155, miR-663b, miR-3648, and miR-6879 are decreased. Mi-27a facilitates the removal of cyclobutane pyrimidine dimers and reduces cell apoptosis [120]. The overexpression of miR-1246 directly upregulates UVB-induced apoptosis through the suppression of rhotekin 2 expression [121]. MiR-34a, miR-134, and miR-383 target the p53 complex via G1 phase arrest and the cell-cycle regulator cyclin, resulting in senescence. Upregulated miR-134 is associated with activation of the NF-κB signaling pathway and SIRT1 inhibition [122]. UVA exposure downregulates miR-155, leading to the upregulation of c-Jun, which affects collagen gene activity in human fibroblasts [123].

### 3.7. Skin Microbiome

The human microbiome is the genome of all microorganisms that reside on or within human tissues, including the skin. The skin hosts a diverse ecosystem, housing up to one million bacteria per square centimeter of surface area, in addition to viruses and fungi [124]. Imbalances of the skin microbiota, an essential element of the skin barrier system, can cause various pathologic conditions, including aging [125,126]. The composition of the skin microbiome varies according to preservation measures, physiological condition, antibacterial treatment, and demographic characteristics [127]. Several studies have used bacterial 16S rRNA gene sequencing to show that the skin microbiome is altered and diversified with aging [128,129,130]. These studies have consistently demonstrated higher alpha diversity in the skin of older than younger people, indicating that the former group may be more susceptible to pathogenic invasion due to altered diversity of the skin microbiota. In addition, during puberty, the density of lipophilic bacteria on the skin increases in line with increasing sebum levels and is lower in the skin of the elderly [130]. Kim et al. [131] studied age-related changes in the skin microbiota of Korean women on the forehead and hands and found that the overall microbial distribution on the forehead varied among age groups, but not on the hands. The authors showed that *Firmicutes* was more abundant in the younger age group, whereas *Bacteroidetes* and *Proteobacteria* increased linearly with aging. The overall microbiome was characterized by beta diversity analysis, and the two groups were significantly different. Zichao et al. [130] recently reported age-related microbiota profiles for both intrinsic skin aging and photoaging. The microbial composition of the elderly group was significantly different from that of the younger group, consistent with previous studies [126]. The authors also conducted a correlation analysis, in which each group showed high enrichment of nine microbial communities (i.e., *Cyanobacteria*, *Staphylococcus*, *Cutibacterium*, *Lactobacillus*, *Corynebacterium*, *Streptococcus*, *Neisseria*, *Candida*, and *Malassezia*) and 18 pathways (e.g., MAPK signaling pathway, glutathione metabolism, photosynthesis, and pantothenate and coenzyme A biosynthesis) that likely contribute to skin aging, suggesting that the skin microbiome plays a key role in skin aging [126]. Various studies related to molecular mechanisms of skin aging are described in Table 2.

## 4. Other Environmental Stressors Associated with Skin Aging

### 4.1. Tobacco Smoke

In addition to UV radiation exposure, tobacco smoke can lead to extrinsic skin aging. Greg et al. [136] recently reported that smoking is associated with more facial lines and decreased face volume, indicative of premature skin aging. Long-term changes in the skin can lead to “smoker’s face”, which is characterized by a grayish appearance and periorbital and perioral facial lines caused by post-inflammatory hyperpigmentation and the breakdown of collagen and elastin fibers in the dermis [131]. One study showed that MMP1 mRNA was significantly increased in the dermis of smokers compared with non-smokers, leading to collagen and elastin fiber breakdown [136]. To better understand the pathogenic role of tobacco in skin aging, Morita et al. [137] performed an in vivo study using a mouse model. They applied a water-soluble tobacco smoke extract to the back skin of mice, which resulted in the loss of normal collagen structure and a significant increase of collagen breakdown in the papillary and upper dermis, mimicking aged skin. Several other studies have suggested a potential pathologic mechanism by which tobacco smoke accelerates skin aging. Tobacco smoke extract decreases procollagen type-I and -III, and increases MMP1 and MMP3, which results in degradation of the ECM structure and abnormal regulation of ECM deposition in human cultured skin fibroblasts [137]. Another study found that tobacco smoke extract is associated with decreased responsiveness to the TGF-β signaling pathway and reduced expression of the TGF-β receptor in the supernatants of cultured skin fibroblasts, leading to decreased synthesis of ECM molecules and structure [138] Additionally, tobacco smoke is a major source of polycyclic aromatic hydrocarbon exposure in humans. Tobacco smoke extract increases the MMP-1 level by activating the aryl hydrocarbon receptor (AhR) signaling pathway in human keratinocytes and fibroblasts [139].

### 4.2. Environmental Pollutants

Exposure to indoor or outdoor chemical, physical, or biological air pollution is one of the major environmental health issues worldwide [140]. It is also associated with an increased risk of extrinsic skin aging, particularly in the form of pigmented spots and wrinkles in white people of European ancestry [141]. Polycyclic aromatic hydrocarbons, which are also present in tobacco extract, are a major effector, triggering the AhR signaling pathway. Due to the high lipophilicity of polycyclic aromatic hydrocarbons, they can penetrate the skin barrier. Activation of AhR pathways lead to increased MMP1 expression in keratinocytes and melanogenesis, via increased tyrosinase enzyme activity in melanocytes [142]. Regarding particulate matter, soot, and nitrogen dioxide, which have emerged as serious pollutants, are well known factors in extrinsic skin aging. Topical exposure of human skin to environmentally relevant concentrations of an internationally established reference standard diesel exhaust mixture increased dark pigmentation of the skin. This tanning effect is due to increased melanogenesis caused by the oxidative stress associated with activation of the p53 signaling pathway [143]. Recently, Li et al. [59] showed that indoor air pollution associated with cooking with solid fuels was significantly associated with 5–8% deeper facial wrinkles and folds, and an increased risk of developing fine wrinkles on the dorsum of the hands in Chinese women. It is likely that indoor combustion of solid fuels activates the same molecular pathways in skin cells as outdoor pollution, and thus causes wrinkles [144].

## 5. Conclusions

Skin aging is a complex process influenced by intrinsic and extrinsic factors. The combined effects of these two aging factors impact the structures and functions of the epidermis and dermis. The most well-known mechanism of skin aging is degradation of ECM due to oxidative stress-induced cellular senescence, resulting in secretion of SASP and activation or inactivation of various signaling pathways. Recently, research on the causes of skin aging has been conducted, and efforts are continuing in research areas such as the skin microbiome and epigenetics. Graphical illustration of molecular aspects of skin aging are described in Figure 3. A better understanding of the molecular mechanisms of skin aging might facilitate the development of novel therapeutic strategies.

## Figures and Tables

**Figure 1 ijms-22-12489-f001:**
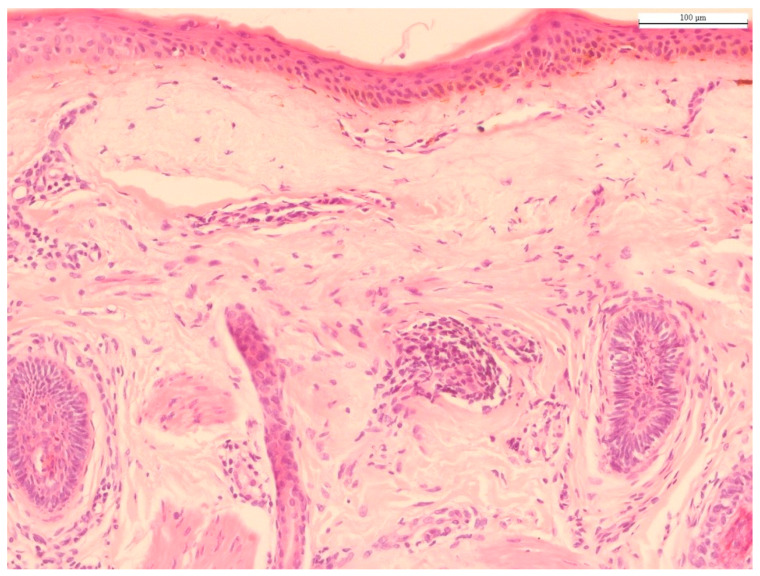
Histology of photoaged skin. The predominant histological finding of photodamaged skin is solar elastosis, i.e., basophilic degeneration of elastotic fibers in the dermis. Solar elastosis is separated from the epidermis by a narrow band of normal-appearing collagen (grenz zone), with collagen fibers arranged horizontally (hematoxylin and eosin staining; original magnification 100×). Source: HJ Lee and M Kim.

**Figure 2 ijms-22-12489-f002:**
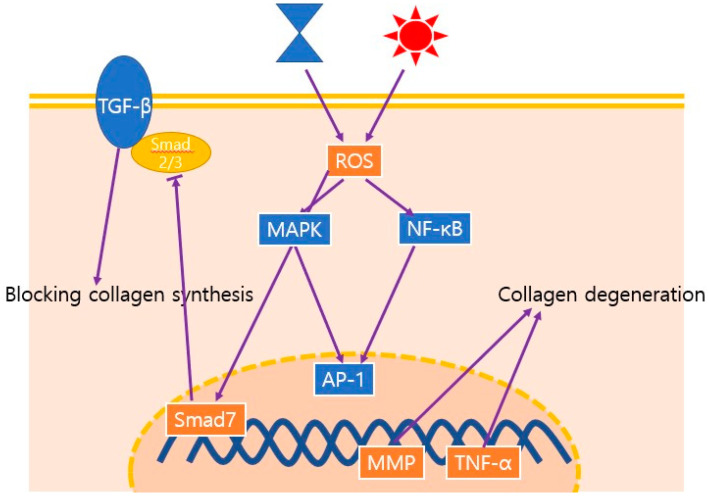
Schematic illustration of the major signaling pathways involved in the aging process; these pathways decrease the density, thickness, and organization of collagen. Both chronological aging and photoaging induce ROS, which leads to the upregulation of MMPs. ROS, reactive oxygen species; MAPK, mitogen-activated protein kinase; NF-κB, nuclear factor kappa B ; AP-1, activated protein 1; MMP, matrix metalloprotease; TNF-α, tumor necrosis factor-α; TGF-β, transforming growth factor-β.

**Figure 3 ijms-22-12489-f003:**
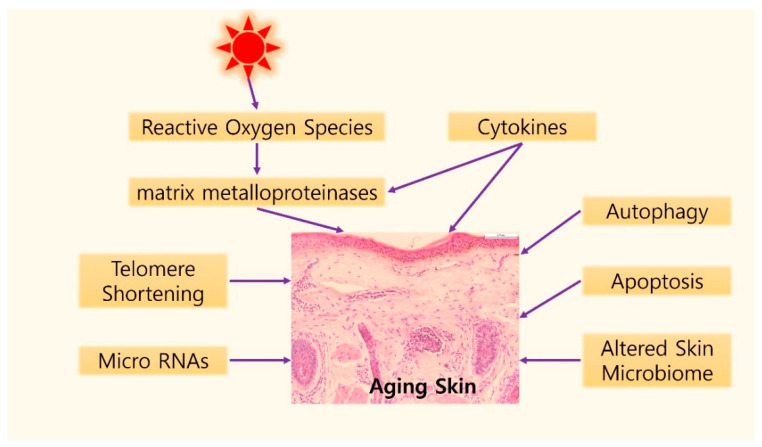
Graphical illustration of possible mechanisms of skin aging. Gradual shortening of telomeres can induce the cellular senescence caused by intrinsic aging. UV exposure induces telomere shortening by producing reactive oxygen species (ROS) in the skin. ROS accumulation results in DNA damage and activation of two major regulatory signaling pathways, the mitogen-activated protein kinase (MAPK) and nuclear factor kappa B (NF-κB) signaling pathways, which induce matrix metalloproteinases activation. In addition, cytokines, such as tumor necrosis factor-α, interleukin (IL)-1, IL-6, IL-18, and Cysteine-rich protein 61 are another important element in the skin aging process. Autophagy and apoptosis might be associated with skin aging. In the field of epigenetics, specific micro RNAs have been reported in aged skin. Finally, studies on the role of the microbiome changed with aging process are being actively conducted.

**Table 1 ijms-22-12489-t001:** Histology of chronological aging and photoaging with supporting studies.

Aging Process	Histological Findings		References
chronological aging	Epidermis	Stratum corneum	[6,7,8]
		Epidermal atrophy	[9,10]
		Langerhans cells	[11]
	Dermis	Fibroblast senescence	[13,14,44]
		Collagen structure	[15,16,17,18,43,44]
		Elastic fibers	[22,23]
Photoaging	Epidermal thickness		[25,26]
	Cell atypism		[27]
	Solar elastosis		[32,34,36,38,40]

**Table 2 ijms-22-12489-t002:** Molecular mechanisms of skin aging. ROS, reactive oxygen species; UV, ultraviolet ray, CCN1; Cellular Communication Network Factor 1.

Molecular Mechanisms	Details	References
Telomere shortening	In intrinsic aging	[48]
	In chronological aging	[49,52]
Oxidative stress and matrix metalloproteinases	ROS and photoaging	[53,55,56]
	Signal cascade activated by ROS	[58,59,61,63,132,133]
Cytokines	Cytokines released by UV radiation	[65,66,67]
	Tumor necrosis factor α	[69]
	Interleukins	[72,73,77,78,79,80,134]
	CCN1	[82,83]
Autophagic control	Association with aging process	[92,93,94,95,96,97,98]
Apoptosis	Association with aging process	[104,107,108,109]
MicroRNAs	Chronological aging	[113,114,115,116,117,118,119,135]
	photoaging	[120,121,122,123]
Skin microbiome	Association with aging process	[128,129,130]

## Data Availability

The data underlying this article will be shared on reasonable request from the corresponding author.

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
