# Peer review of "Structural and Functional Changes and Possible Molecular Mechanisms in Aged Skin"

_ijms, 2021, doi:10.3390/ijms222212489_

Round 1
Reviewer 1 Report
This is a great review about skin aging. This review underlines the histological and molecular changes regulating chronological and photo-aging. Manuscript is presented nicely. I have following few suggestions,
- Some of the sections such as 3.3 (cytokines in skin aging), 3.5 (miRNAs in organisms) and 3.6 skin microbiome are not comprehensive and discussed briefly. Authors should discuss these sections more in details and wherever possible cite more relevant articles.
- Heading for 3.5 miRNAs in organisms could be changed to miRNAs in skin aging or miRNAs.
- A graphical illustration of skin chronological and photo-aging highlighting different molecular factors discussed in the manuscript should be included.
Author Response
Dear. Reviewer 1
Thank you very much for your kind editorial letter.
We have attempted to carefully and thoroughly address all concerns raised by the editors and referees. With the help of your suggestions, we believe our manuscript has significantly improved.
The major changes are indicated by red font.
We trust that we therewith have fulfilled all the Editor’s and Reviewer’s requests.
Thank you very much for your consideration.
This is a great review about skin aging. This review underlines the histological and molecular changes regulating chronological and photo-aging. Manuscript is presented nicely. I have following few suggestions,
Some of the sections such as 3.3 (cytokines in skin aging), 3.5 (miRNAs in organisms) and 3.6 skin microbiome are not comprehensive and discussed briefly. Authors should discuss these sections more in details and wherever possible cite more relevant articles.
 Thank you for kind comment. I added more discussions in each parts.
Heading for 3.5 miRNAs in organisms could be changed to miRNAs in skin aging or miRNAs.
 Thank you for the comment. I revised them.
A graphical illustration of skin chronological and photo-aging highlighting different molecular factors discussed in the manuscript should be included.
 Thank you for the comment. I added figure 3.
Thank you again for your helpful review of our article.
Sincerely,
Miri Kim, MD. PhD
Department of Dermatology, Yeouido St. Mary’s Hospital, College of Medicine, The Catholic University of Korea, #10, 63-ro, Yeongdeungpo-gu, Seoul 07345, Korea
E-mail: [email protected]
Reviewer 2 Report
In this review article: Structural and functional changes and possible molecular mechanisms in aged skin, the authors discussed the various mechanisms involved in the aged skin, which have clinical importance.
The review is interesting, but needs improvements:
1- The source of Fig.1 showing the aged skin should be mentioned. Also, if possible add a Fig. Showing normal skin for comparison. If you have any TEM figures showing the aged skin, it will be great to add it to light microscopic figures.
2-The role of apoptosis in the pathology of aging skin should be added or clarified
3-The authors should insert up to date tables showing various studies related to their study; specifically related to autophagy, apoptosis and other mechanisms mentioned in their study.
4-The authors discussed the role of autophagy in the aged skin, but it is better to add a short paragraph discussing the various molecular mechanisms of autophagy pathway. I recommend the authors read and cite this article:Klinsky et al. Guidelines for the use and interpretation of assays for monitoring autophagy (4th edition)1. Autophagy. 2021 Jan;17(1):1-382. doi: 10.1080/15548627.2020.1797280. Epub 2021 Feb 8. PMID: 33634751; PMCID: PMC7996087.
5- The manuscript should be revised and checked by native English speakers.
Author Response
Dear. Reviewer 2
Thank you very much for your kind editorial letter.
We have attempted to carefully and thoroughly address all concerns raised by the editors and referees. With the help of your suggestions, we believe our manuscript has significantly improved.
The major changes are indicated by red font.
We trust that we therewith have fulfilled all the Editor’s and Reviewer’s requests.
Thank you very much for your consideration.
1- The source of Fig.1 showing the aged skin should be mentioned. Also, if possible add a Fig. Showing normal skin for comparison. If you have any TEM figures showing the aged skin, it will be great to add it to light microscopic figures.
 Thank you for kind comment. I added the source of Figure 1. However, we cannot find normal skin histology or TEM figures in our database.
2- The role of apoptosis in the pathology of aging skin should be added or clarified
 Thank you for the comment. I added section 3.5 for apoptosis in skin aging.
3- The authors should insert up to date tables showing various studies related to their study; specifically related to autophagy, apoptosis and other mechanisms mentioned in their study.
 Thank you for the comment. I added Table 1 and Table 2 for the data tables.
4- The authors discussed the role of autophagy in the aged skin, but it is better to add a short paragraph discussing the various molecular mechanisms of autophagy pathway. I recommend the authors read and cite this article:Klinsky et al. Guidelines for the use and interpretation of assays for monitoring autophagy (4th edition)1. Autophagy. 2021 Jan;17(1):1-382. doi: 10.1080/15548627.2020.1797280. Epub 2021 Feb 8. PMID: 33634751; PMCID: PMC7996087.
 Thank you for kind comment. I added the references in section 3.4
5- The manuscript should be revised and checked by native English speakers.
 Thank you for the comment. This manuscript underwent additional English proofreading.
The English in this document has been checked by at least two professional editors, both native speakers of English. For a certificate, please see:
http://www.textcheck.com/certificate/t1LGZM
Thank you again for your helpful review of our article.
Sincerely,
Miri Kim, MD. PhD
Department of Dermatology, Yeouido St. Mary’s Hospital, College of Medicine, The Catholic University of Korea, #10, 63-ro, Yeongdeungpo-gu, Seoul 07345, Korea
E-mail: [email protected]
Reviewer 3 Report
- Define abbreviations at the first occurence in the text, and use the abbreviation threrafter.
Abstract:
1st sentence " ... estradiol on chondrocyte ... " --> ... estradiol (ED) on chondrocyte ...
"Results: Estradiol (ED) decreased ..." -->Results: ED decreased ...
"Conclusion: Our results show that estradiol ..." --> Conclusion: Our results show that ED ...
Similar problems with abbreviations are found throughout the manuscript.
2. 2.1 Animals: "... average body weight of 20-24 g ..." Was this a range rather than an average?
3. 2.2 Model ... Add a definition of collagenase units.
4. 2.4 Primary Hondrocyte ..." --> 2.4 Primary Condrocyte ...
5. Table 1: What are the units?
6. "Women after the age of 60 are predominantly affected representing 18% of OA patients compared to males representing 10% [1]." Does this mean that 72% of OA patients are under the age of 60?
7. It is suggested that the authors get help with the English language in revising this manuscript.
Author Response
Thank you very much for your kind editorial letter.
Sincerely,
Miri Kim, MD. PhD
Department of Dermatology, Yeouido St. Mary’s Hospital, College of Medicine, The Catholic University of Korea, #10, 63-ro, Yeongdeungpo-gu, Seoul 07345, Korea
E-mail: [email protected]
Round 2
Reviewer 2 Report
The manuscript is improved, but Fig.3 has mistakes in language such as autophage!! Please change to autophagy. Also, this figure should be improved as the title of the figure is the molecular mechanisms of aged skin, but no any molecules of autophagy or apoptosis are shown in the figure.
Author Response
Dear. Reviewer 2
Thank you very much for your kind editorial letter.
We have attempted to carefully and thoroughly address all concerns raised by the editors and referees. With the help of your suggestions, we believe our manuscript has significantly improved.
The major changes are indicated by red font.
We trust that we therewith have fulfilled all the Editor’s and Reviewer’s requests.
Thank you very much for your consideration.
As we tried to express as many possible mechanisms as possible about skin aging in one graphical illustration, we could not include the details of each part. We ask for your understanding. I revised "molecular mechanisms" to possible mechanism in line 486, and we corrected the typo in the figure 3.
Thank you again for your helpful review of our article.
Sincerely,
Miri Kim, MD. PhD